# A REDUCTION APPROACH TO CONSTRAINED REINFORCEMENT LEARNING

## ABSTRACT

Many applications of reinforcement learning (RL) optimize a long-term reward subject to risk, safety, budget, diversity or other constraints. Though constrained RL problem has been studied to incorporate various constraints, existing methods either tie to specific families of RL algorithms or require storing infinitely many individual policies found by an RL oracle to approach a feasible solution. In this paper, we present a novel reduction approach for constrained RL problem that ensures convergence when using any off-the-shelf RL algorithm to construct an RL oracle yet requires storing at most constantly many policies. The key idea is to reduce the constrained RL problem to a distance minimization problem, and a novel variant of Frank-Wolfe algorithm is proposed for this task. Throughout the learning process, our method maintains at most constantly many individual policies, where the constant is shown to be worst-case optimal to ensure convergence of any RL oracle. Our method comes with rigorous convergence and complexity analysis, and does not introduce any extra hyper-parameter. Experiments on a grid-world navigation task demonstrate the efficiency of our method.

## 1 INTRODUCTION

Contemporary approaches in reinforcement learning (RL) largely focus on optimizing the behavior of an agent against a single reward function. RL algorithms like value function methods (Zou et al., 2019; Zheng et al., 2018) or policy optimization methods (Chen et al., 2019; Zhao et al., 2017) are widely used in real-world tasks. This can be sufficient for simple tasks. However, for complicated applications, designing a reward function that implicitly defines the desired behavior can be challenging. For instance, applications concerning risk (Geibel & Wysotzki, 2005; Chow & Ghavamzadeh, 2014; Chow et al., 2017), safety (Chow et al., 2018) or budget (Boutilier & Lu, 2016; Xiao et al., 2019) are naturally modelled by augmenting the RL problem with orthant constraints. Exploration suggestions, such as to visit all states as evenly as possible, can be modelled by using a vector to measure the behavior of the agent, and to find a policy whose measurement vector lies in a convex set (Miryoosefi et al., 2019).

To solve RL problem under constraints, existing methods either ensure convergence only on a specific family of RL algorithms, or treat the underlying RL algorithms as a black box oracle to find individual policy, and look for mixed policy that randomizes among these individual policies. Though the second group of methods has the advantage of working with arbitrary RL algorithms that best suit the underlying problem, existing methods have practically infeasible memory requirement. To get an $\epsilon$-approximate solution, they require storing $O(1/\epsilon)$ individual policies, and an exact solution requires storing infinitely many policies. This limits the prevalence of such methods, especially when the individual policy uses deep neural networks.

In this paper, we propose a novel reduction approach for the general convex constrained RL (C2RL) problem. Our approach has the advantage of the second group of methods, yet requires storing at most constantly many policies. For a vector-valued Markov Decision Process (MDP) and any given target convex set, our method finds a mixed policy whose measurement vector lies in the target convex set, using any off-the-shelf RL algorithm that optimizes a scalar reward as a *RL oracle*. To do so, the C2RL problem is reduced to a distance minimization problem between a polytope and a convex set, and a novel variant of Frank-Wolfe type algorithm is proposed to solve this distance minimization problem. To find an $\epsilon$-approximate solution in an $m$-dimensional vector-valued MDP,

Table 1: Comparison with previous approaches. To find an $\epsilon$-approximate solution, time complexity under orthant or convex constraints is compared using the numbers of RL oracle calls. The memory requirement is measured by the number of individual policies stored for an $\epsilon$-approximate solution.

| Method | Orthant constraint | Convex constraint | Converge for any RL algo. | No extra hyper-parameter | Memory requirement |
|---|---|---|---|---|---|
| Tessler et al. (2018) | To a fixed point | ✗ | ✗ | ✗ | 1 |
| Le et al. (2019) | $O(1/\epsilon)$ | ✗ | ✓ | ✗ | $O(1/\epsilon)$ |
| Miryoosefi et al. (2019) | $O(1/\epsilon)$ | $O(1/\epsilon)$ | ✓ | ✗ | $O(1/\epsilon)$ |
| C2RL (this paper) | $O(1/\epsilon)$ | $O(1/\epsilon)$ | ✓ | ✓ | $\leq m+1$ |

our method only stores at most $m + 1$ policies, which improves from infinitely many $O(1/\epsilon)$ (Le et al., 2019; Miryoosefi et al., 2019) to a constant. We also show this $m + 1$ constant is worst-case optimal to ensure convergence of RL algorithms using deterministic policies. Moreover, our method introduces no extra hyper-parameter, which is favorable for practical usage. A preliminary experimental comparison demonstrates the performance of the proposed method and the sparsity of the policy found.

## 2 RELATED WORK

For high dimensional constrained RL, one line of approaches incorporates the constraint as a penalty signal into the reward function, and makes updates in a multiple time-scale scheme (Tessler et al., 2018; Chow & Ghavamzadeh, 2014). When used with policy gradient or actor-critic algorithms (Sutton & Barto, 2018), this penalty signal guides the policy to converge to a constraint satisfying one (Paternain et al., 2019; Chow et al., 2017). However, the convergence guarantee requires the RL algorithm can find a single policy that satisfies the constraint, hence ruling out methods that search for deterministic policies, such as Deep Q-Networks (DQN) (Mnih et al., 2013), Deep Deterministic Policy Gradient (DDPG) (Lillicrap et al., 2015) and their variants (Van Hasselt et al., 2015; Wang et al., 2016; Fujimoto et al., 2018; Barth-Maron et al., 2018).

Another line of approaches uses a game-theoretic framework, and does not tie to specific families of RL algorithm. The constrained problem is relaxed to a zero-sum game, whose equilibrium is solved by online learning (Agarwal et al., 2018). The game is played repeatedly, each time any RL algorithm can be used to find a *best response* policy to play against a no-regret online learner. The *mixed policy* that uniformly distributed among all played policies can be shown to converge to an optimal policy of the constrained problem (Freund & Schapire, 1999; Abernethy et al., 2011). Taking this approach, Le et al. (2019) uses Lagrangian relaxation to solve the orthant constraint case, and Miryoosefi et al. (2019) uses conic duality to solve the convex constraint case. However, since the convergence is established by the no-regret property, the policy found by these methods requires randomization among policies found during the learning process, which limits their prevalence.

Different from the game-theoretic approaches, we reduce the C2RL to a distance minimization problem and propose a novel variant of Frank-Wolfe (FW) algorithm to solve it. Our result builds on recent finding that the standard FW algorithm emerges as computing the equilibrium of a special convex-convave zero sum game (Abernethy & Wang, 2017). This connects our approach with previous approaches from game-theoretic framework (Agarwal et al., 2018; Le et al., 2019; Miryoosefi et al., 2019). The main advantage of our reduction approach is that the convergence of FW algorithm does not rely on the no-regret property of an online learner. Hence there is no need to introduce extra hyper-parameters, such as learning rate of the online learner, and intuitively, we can eliminate unnecessary policies to achieve better sparsity. To do so, we extend Wolfe's method for minimum norm point problem (Wolfe, 1976) to solve our distance minimization problem. Throughout the learning process, we maintain an active policy set, and constantly eliminate policies whose measurement vector are affinely dependent of others. Unlike norm function in Wolfe's method, our objective function is not strongly convex. Hence we cannot achieve the linear convergence of Wolfe's method as shown in Lacoste-Julien & Jaggi (2015). Instead, we analyze the complexity of our method based on techniques from Chakrabarty et al. (2014). A theoretical comparison between our method and various approaches in constrained RL is provided in Table 1.

## 3 PRELIMINARIES

A *vector-valued Markov decision process* can be identified by a tuple $\{\mathcal{S}, \mathcal{A}, \beta, P, \boldsymbol{c}\}$, where $\mathcal{S}$ is a set of states, $\mathcal{A}$ is the set of actions and $\beta$ is the initial state distribution. At the start of each episode, an initial state $s_0$ is drawn following the distribution $\beta$. Then, at each step $t = 0, 1, \dots$, the agent observes a state $s_t \in \mathcal{S}$ and makes a decision to take an action $a_t$. After $a_t$ is chosen, at the next observation the state evolves to state $s_{t+1} \in \mathcal{S}$ with probability $P(s_{t+1}|s_t, a_t)$. However, instead of a scalar reward, in our setting, the agent receives an $m$-dimensional vector $\boldsymbol{c}_t \in \mathbb{R}^m$ that may implicitly contain measurements of reward, risk or violation of other constraints. The episode ends after a certain number of steps, called the horizon, or when a terminate state is reached.

Actions are typically selected according to a policy $\pi$, where $\pi(s)$ is a distribution over actions for any $s \in \mathcal{S}$. Policies that take a single action for any state are *deterministic policies*, and can be identified by the mapping $\pi : \mathcal{S} \mapsto \mathcal{A}$. The set of all deterministic policies is denoted by $\Pi$. For a discount factor $\gamma \in [0, 1)$, the discounted long-term *measurement vector* of a policy $\pi \in \Pi$ is defined as

$$\boldsymbol{c}(\pi) := \mathbb{E}\Big(\sum_{t=0}^{T} \gamma^t \boldsymbol{c}_t(s_t, \pi(s_t))\Big), \tag{1}$$

where the expectation is over trajectories generated by the described random process.

Unlike unconstrained setting, for a constrained RL problem, it is possible that all feasible policies are non-deterministic (see Appendix D for an example). This limits the usage of RL algorithms that search for deterministic policies in the setting of constrained RL problem.

One workaround is to use mixed policies. For a set of policies $\mathcal{U}$, a mixed policy is a distribution over $\mathcal{U}$, and the set of all mixed policies over $\mathcal{U}$ is denoted by $\Delta(\mathcal{U})$. To execute a mixed policy $\mu \in \Delta(\mathcal{U})$, we first select a policy $\pi \in \mathcal{U}$ according to $\pi \sim \mu(\pi)$, and then execute $\pi$ for the entire episode. Altman (1999) shows that any $\boldsymbol{c}(\cdot)$ achievable can be achieved by some *mixed deterministic policies* $\mu \in \Delta(\Pi)$. Therefore, though an off-shelves RL algorithm may not converge to any constraint-satisfying policy, it can be used as a subroutine to find individual policies (possibly deterministic), and a randomization among these policies can converge to a feasible policy. The discounted long-term measurement vector of a mixed policy $\mu \in \Delta(\Pi)$ is defined similarly

$$\boldsymbol{c}(\mu) := \mathbb{E}_{\pi \sim \mu}(\boldsymbol{c}(\pi)) = \sum_{\pi \in \Pi} \mu(\pi)\boldsymbol{c}(\pi). \tag{2}$$

For a mixed policy $\mu \in \Delta(\mathcal{U})$, its *active set* is defined to be the set of policies with non-zero weights $\mathcal{A} := \{\pi \in \mathcal{U}|\mu(\pi) > 0\}$. The memory requirement of storing $\mu$, is then proportional to the size of its active set. Since a mixed policy can be interpreted as a convex combination of policies in its active set, in the following, the term *sparsity* of a mixed policy refers to the sparsity of this combination.

Our learning problem, the convex constrained reinforcement learning (C2RL), is to find a policy whose expected long-term measurement vector lies in a given convex set; i.e., for a given convex target set $\mathcal{C} \subset \mathbb{R}^m$, our target is to

$$\text{find } \mu^* \text{ such that } \boldsymbol{c}(\mu^*) \in \Omega \qquad \text{(C2RL)}. \tag{3}$$

Any policy $\mu^*$ that satisfies $\boldsymbol{c}(\mu^*) \in \Omega$ is called a *feasible policy*, and a C2RL problem is *feasible* if there exists some feasible policies. In the following, we assume the C2RL problem is feasible.

## 4 APPROACH, ALGORITHM AND ANALYSIS

We now show how the C2RL (3) can be reduced to a distance minimization problem (7) between a polytope and a convex set. A novel variant of Frank-Wolfe-type algorithm is then proposed to solve the distance minimization problem, followed by theoretic analysis about convergence and sparsity of the proposed method.

### 4.1 REDUCE C2RL TO A DISTANCE MINIMIZATION PROBLEM

Let $||\cdot||$ denote the Euclidean norm. For a convex set $\Omega \in \mathbb{R}^m$, let $\texttt{Proj}_\Omega(\boldsymbol{x}) \in \arg\min_{\boldsymbol{y} \in \Omega} ||\boldsymbol{x} - \boldsymbol{y}||$ be the projection operator, and $\texttt{dist}^2(\boldsymbol{x}, \Omega) := \frac{1}{2}||\boldsymbol{x} - \texttt{Proj}_\Omega(\boldsymbol{x})||^2$ be half of the squared Euclidean

distance function. Then we consider the problem to find a policy whose measurement vector is closest to the target convex set,

$$\arg\min_{\mu \in \Delta(\Pi)} \texttt{dist}^2(\boldsymbol{c}(\mu), \Omega). \tag{4}$$

A policy $\mu^* \in \Delta(\Pi)$ is defined to be an *optimal* solution if it minimizes (4). Otherwise, the *approximation error* of $\mu \in \Delta(\Pi)$ is defined as

$$\texttt{err}(\mu) := \texttt{dist}^2(\boldsymbol{c}(\mu), \Omega) - \texttt{dist}^2(\boldsymbol{c}(\mu^*), \Omega) \qquad \text{(Approximation Error)} \tag{5}$$

where $\mu^*$ is an optimal solution, and a policy is defined to be an $\epsilon$-approximate solution if its approximation error is no larger than $\epsilon$.

When C2RL (3) is feasible, the equivalence of being optimal to (4) and being feasible to C2RL can be easily established. Since a feasible policy of C2RL problem lies inside $\Omega$, it minimizes the non-negative $\texttt{dist}^2$ function, and hence is optimal to (4). Vice versa, any optimal solution to (4) lies inside $\Omega$ and is a feasible solution to C2RL.

From a geometric perspective, let $\boldsymbol{c}(\Pi) := \{\boldsymbol{c}(\pi)|\pi \in \Pi\}$ be the set of all values achievable by deterministic policies. If the MDP has finite states and actions (though may be extremely large), then $\Pi$ is finite as well, and hence $\boldsymbol{c}(\Pi)$ contains finitely many points in $\mathbb{R}^m$. Then the set of values achievable by mixed deterministic policies

$$\boldsymbol{c}(\Delta(\Pi)) := \{\boldsymbol{c}(\mu)|\mu \in \Delta(\Pi)\} = \{\sum_\pi \mu(\pi)\boldsymbol{c}(\pi) \mid \sum_\pi \mu(\pi) = 1, \mu(\pi) \geq 0\} \subset \mathbb{R}^m \tag{6}$$

is the convex hull of $\boldsymbol{c}(\Pi)$; i.e., $\boldsymbol{c}(\Delta(\Pi))$ is a $m$-dimension polytope whose vertices are $\boldsymbol{c}(\Pi)$. Therefore finding a policy whose value is closest to the target convex set (4) is equivalent to find a point in the polytope $\boldsymbol{c}(\Delta(\Pi))$ that is closest to the convex set $\Omega$

$$\arg\min_{\boldsymbol{c}(\mu) \in \boldsymbol{c}(\Delta(\Pi))} \texttt{dist}^2(\boldsymbol{c}(\mu), \Omega) \qquad \text{(Distance minimization problem)}. \tag{7}$$

To solve this constrained optimization problem, it might be tempting to consider projection methods. However, constructing a projection operator for $\boldsymbol{c}(\Delta(\Pi))$ is non-trivial. For any given measurement vector, it is obscure how to modify a general RL algorithm to update the parameters such that the discounted expected measurement vector is closest to the given value. Therefore, projection-free methods are preferable for this task.

Frank-Wolfe (FW) algorithm does not require any projection operation, instead it uses a linear minimizer oracle. Intuitively, finding a linear minimizer is similar to the reward maximization process of what a general RL algorithm does. In section 4.3, we formalize this idea. We show that after simple modifications, any RL algorithm that maximizes a scalar reward can be used to construct such a linear minimizer oracle. Before getting into details of the construction process, we discuss FW-type algorithms over polytope and its applications in the distance minimization problem (7).

## 4.2 Distance Minimization by Frank-Wolfe-type Algorithms

The Frank-Wolfe algorithm (FW) is a first-order method to minimize a convex function $f : \mathcal{P} \mapsto \mathbb{R}$ over a compact and convex set $\mathcal{P}$, with only access to a *linear minimizer oracle*. When the feasible set is a polytope $\mathcal{P} := \texttt{conv}(\{\boldsymbol{s}_1, \boldsymbol{s}_2, \ldots, \boldsymbol{s}_n\}) \subset \mathbb{R}^m$ defined as the convex hull of finitely many points, FW-type algorithms are discussed by Lacoste-Julien & Jaggi (2015) to optimize

$$\min_{\boldsymbol{x} \in \mathcal{P}} f(\boldsymbol{x}) \quad \text{using} \quad \texttt{Oracle}(\text{v}) := \arg\min_{\boldsymbol{s} \in \{\boldsymbol{s}_1, \ldots, \boldsymbol{s}_n\}} \boldsymbol{s}^T \boldsymbol{v}. \tag{8}$$

The standard FW (Algorithm 2 in Appendix A.1) consists of making repeated calls to the linear minimizer oracle to find an improving point $\boldsymbol{s}$, followed by a convex averaging step of the current iterate $\boldsymbol{x}_{t-1}$ and the oracle's output $\boldsymbol{s}$.

If we have already constructed a $\texttt{RL\_oracle}(\boldsymbol{\lambda})$ that outputs a policy $\pi \in \arg\min_{\pi \in \Pi} \boldsymbol{\lambda}^T \boldsymbol{c}(\pi)$ together with its measurement vector $\boldsymbol{c}(\pi)$, then the distance minimizing problem (7) can be solved with standard FW by using

$$\pi, \boldsymbol{c}(\pi) \leftarrow \texttt{RL\_oracle}(\nabla \texttt{dist}^2(\boldsymbol{x}_{t-1}, \Omega)) = \texttt{RL\_oracle}(\boldsymbol{x}_{t-1} - \texttt{Proj}_\Omega(\boldsymbol{x}_{t-1})) \tag{9}$$

---

**Algorithm 1** Convex Constrained Reinforcement Learning (C2RL)

---

**Input.** $\texttt{RL\_Oracle}$ constructed by any RL algorithm, projection operator to target set $\texttt{Proj}_\Omega$.
**Initialize.** Random policy $\pi$, value $\boldsymbol{x} = \boldsymbol{c}(\pi)$, active sets $\mathcal{S}_p := [\pi], \mathcal{S}_c := [\boldsymbol{x}]$ and weight $\boldsymbol{\lambda} = [1]$.
**Output.** Mixed policy $\mu$ and its value $\boldsymbol{c}(\mu)$ s.t. $\boldsymbol{c}(\mu)$ minimizes the distance to the target set $\Omega$.

 1: **while** true **do**                                                                                          // Major cycle
 2:     $\quad \boldsymbol{\omega} \leftarrow \texttt{Proj}_\Omega(\boldsymbol{x})$
 3:     $\quad \pi, \boldsymbol{c}(\pi) \leftarrow \texttt{RL\_Oracle}(\boldsymbol{x} - \boldsymbol{\omega})$                                       // Potential improving point
 4:     $\quad$ **if** $(\boldsymbol{x} - \boldsymbol{\omega})^T(\boldsymbol{x} - \boldsymbol{c}(\pi)) \leq \epsilon$ **then** break
 5:     $\quad$ **if** $\mathcal{S}_c \cup \{\boldsymbol{c}(\pi)\}$ is affinely independent **then** $\mathcal{S}_c \leftarrow \mathcal{S}_c \cup \{\boldsymbol{c}(\pi)\}, \mathcal{S}_p \leftarrow \mathcal{S}_p \cup \{\pi\}$
 6:     $\quad$ **while** true **do**                                                                                  // Minor cycle
 7:         $\quad\quad \boldsymbol{y}, \boldsymbol{\alpha} \leftarrow \texttt{AffineMinimizer}(\mathcal{S}_c, \boldsymbol{\omega})$             // $\boldsymbol{y} = \arg\min_{\boldsymbol{s} \in \texttt{aff}(\mathcal{S}_c)} ||\boldsymbol{s} - \boldsymbol{\omega}||_2$
 8:         $\quad\quad$ **if** $\alpha_{\boldsymbol{s}} > 0$ for all $\boldsymbol{s}$ **then** break                                     // $\boldsymbol{y} \in \texttt{conv}(\mathcal{S}_c)$
 9:         $\quad\quad$ // If $\boldsymbol{y} \notin \texttt{conv}(\mathcal{S}_c)$, then update $\boldsymbol{y}$ to the intersection of $\texttt{conv}(\mathcal{S}_c)$ and segment joining $\boldsymbol{x}$
                 $\quad\quad$ and $\boldsymbol{y}$. Then remove points in $\mathcal{S}_c$ unnecessary for describing $\boldsymbol{y}$.
10:         $\quad\quad \theta \leftarrow \min_{i:\alpha_i \leq 0} \frac{\lambda_i}{\lambda_i - \alpha_i}$                                   // Recall $\boldsymbol{\lambda}$ satisfies $\boldsymbol{x} = \sum_{\boldsymbol{s} \in \mathcal{S}_c} \lambda_{\boldsymbol{s}} \boldsymbol{s}$
11:         $\quad\quad \boldsymbol{y} \leftarrow \theta\boldsymbol{y} + (1 - \theta)\boldsymbol{x}, \lambda_i = \theta\alpha_i + (1 - \theta)\lambda_i$
12:         $\quad\quad \mathcal{S}_c \leftarrow \{\boldsymbol{c}(\pi_i) | \boldsymbol{c}(\pi_i) \in \mathcal{S}_c \text{ and } \lambda_i > 0\}, \mathcal{S}_p \leftarrow \{\pi_i | \pi_i \in \mathcal{S}_p \text{ and } \lambda_i > 0\}$
13:     $\quad$ **end while**
14:     $\quad$ Update $\mu \leftarrow \sum_{\pi \in \mathcal{S}_p} \lambda_\pi \pi, \boldsymbol{x} \leftarrow \boldsymbol{y}, \boldsymbol{\lambda} \leftarrow \boldsymbol{\alpha}$.
15: **end while**
16: **return** $\mu, \boldsymbol{c}(\mu) \leftarrow \boldsymbol{x}$

---

to find an improving policy and its measurement vector. For $\eta_t := \frac{2}{t+2}$, the convex averaging steps

$$\mu_t \leftarrow (1 - \eta_t)\mu_{t-1} + \eta_t\pi, \quad \boldsymbol{x}_t \leftarrow (1 - \eta_t)\boldsymbol{x}_{t-1} + \eta_t\boldsymbol{c}(\pi), \tag{10}$$

then maintain the mixed policy, and the corresponding measurement vector, respectively.

However, after $T$ rounds of iteration, the $\mu_t$ found has an active set containing up to $T$ individual polices, and is not sparse enough. If neural networks are used to parameterize the policy, that requires storing $T$ copies of parameters for the individual network, which is unaffordable for large-scale usage.

To find even more sparse policies, we turn to variants of FW-type algorithms. In particular, Wolfe's method for minimum norm point in a polytope (Wolfe, 1976; De Loera et al., 2018). In Wolfe's method (Algorithm 3 in Appendix A.2), the loop in FW is called a *major cycle*, and the convex averaging step is replaced by a weight optimization process, called *minor cycle*. Wolfe's method maintains an *active set* $\mathcal{S}$, and the current point can be represented by a sparse combination of points in the active set. The minor cycles maintain $\mathcal{S}$ to be an affinely independent set such that the affine minimizer is inside $\mathcal{S}^t$, which Wolfe calls *corrals*. Recall an affine minimizer is defined as $\arg\min_{\boldsymbol{s} \in \texttt{aff}(\mathcal{S})} ||\boldsymbol{s}||_2$, where $\texttt{aff}(\mathcal{S}) := \{\boldsymbol{y} | \boldsymbol{y} = \sum_{\boldsymbol{z} \in \mathcal{S}} \alpha_{\boldsymbol{z}}^T \boldsymbol{x}, \sum_{\boldsymbol{z} \in \mathcal{S}} \alpha_{\boldsymbol{z}} = 1\}$ is the affine hull formed by $\mathcal{S}$. Since the active set is affinely independent, the number of active atoms is at most $m + 1$ at any time. Wolfe's method is shown to strictly decrease the approximation error between two major cycles.

### 4.3 OUR MAIN ALGORITHM

The main obstacle to apply Wolfe's method to our distance minimization problem (7) is that the objective function in Wolfe's method is the norm function. However, in our problem, the objective function is the distance function to a convex set. Unlike the norm function, the distance function to a convex set is not strongly convex and affine minimizer is ill-defined with respect to a convex set. To tackle these problems, we modify the Wolfe's method. At the core of our new variant of FW algorithm, we add a projection step to Wolfe's method.

**Projection Step** In each major cycle, we minimize the distance to a projected point $\boldsymbol{\omega} := \texttt{Proj}_\Omega(\boldsymbol{x})$. Intuitively, since the distance to the convex set is upper bounded by the distance to this projected point $\boldsymbol{\omega}$, if the distance to $\boldsymbol{\omega}$ converges, so does the distance to the target convex set.

Formally, for a set of points $\mathcal{S} \subset \mathbb{R}^m$, and a point $\boldsymbol{x} \in \mathbb{R}^m$, we extend the definition of an affine minimizer to define *affine minimizer with respect to* $\boldsymbol{x}$ as $\arg\min_{\boldsymbol{s} \in \texttt{aff}(\mathcal{S})} ||\boldsymbol{s} - \boldsymbol{x}||_2$. For $\boldsymbol{x}$ being

the affine minimizer of $\mathcal{S}$ with respect to $\boldsymbol{\omega}$, the extended affine minimizer property gives

$$\text{Given } \omega, \forall \boldsymbol{v} \in \texttt{aff}(\mathcal{S}), (\boldsymbol{v} - \boldsymbol{x})^T(\boldsymbol{x} - \boldsymbol{\omega}) = 0 \quad \text{(Extended affine minimizer property)} \quad (11)$$

Similar to Wolfe's method, our C2RL method (Algo. 1) contains an outer loop (called major cycle) to find improving policies and their measurement vectors, and an inner loop (called minor cycle) to maintain the affinely independent property of the active set $\mathcal{S}_c$. At the start of each major cycle step, the $\mathcal{S}_c$ is an affinely independent set. Then, the RL_oracle (defined in (15)) finds a potential improving policy $\pi \in \mathcal{U}$, and its long-term measurement vector $\boldsymbol{c}(\pi)$. If the $\boldsymbol{c}(\pi)$ does not get strictly closer to the $\boldsymbol{\omega} := \texttt{Proj}(\boldsymbol{x})$, then we are done, and $\boldsymbol{x}$ is the optimal value. Otherwise, the $\boldsymbol{c}(\pi)$ is added into the active set, and the minor cycle is run to eliminate policies whose measurement vectors are affinely dependent.

Line 6 to line 13 contains the minor cycle, which is the same as the original Wolfe's method (except in line 6, we find affine minimizer with respect to $\boldsymbol{\omega}$). The elimination is executed as a series of affine projections. The minor cycle terminates if active set $\mathcal{S}_c$ is affinely independent. Though the interleaving of major and minor cycles oscillate the size of active set $\mathcal{S}_c$, the minor cycles keep $|\mathcal{S}_c|$ an affinely independent set, and is terminated whenever $\mathcal{S}_c$ contains a single element. Therefore at the start of any major cycle, the size of the active set satisfies $|\mathcal{S}_c| \in [0, m+1]$. More background about the minor cycle in Wolfe's method is provided in Appendix A.2.

**Construction of RL Oracle** The construction of our RL oracle can use any off-the-shelf RL algorithm that maximizes a scalar reward. For any given $\boldsymbol{\lambda} \in \mathbb{R}^m$, we define any algorithm that finds a policy minimizing the linear function $\boldsymbol{\lambda}^T \boldsymbol{c}(\cdot)$ as a *RL oracle*, that is

$$\texttt{RL\_oracle}_\texttt{p}(\boldsymbol{\lambda}) \in \underset{\pi \in \Pi}{\arg\min} \, \boldsymbol{\lambda}^T \boldsymbol{c}(\pi). \quad (12)$$

Recall that standard RL algorithm receives a scalar reward after each state transition, instead of the long-term measurement vector $\boldsymbol{c}(\pi) \in \mathbb{R}^m$. We then use the following linear property to reformulate the right hand side of (12) to a standard RL problem

$$\underset{\pi \in \Pi}{\arg\min} \, \boldsymbol{\lambda}^T \boldsymbol{c}(\pi) = \underset{\pi \in \Pi}{\arg\min} \, \boldsymbol{\lambda}^T \mathbb{E}(\sum_{t=0}^{T} \gamma^t \boldsymbol{c}_t) = -\underset{\pi \in \Pi}{\arg\max} \, \mathbb{E}(\sum_{t=0}^{T} \gamma^t (-\boldsymbol{\lambda}^T \boldsymbol{c}_t)). \quad (13)$$

This shows that if we consider the Markov decision process with the same state, action, and transition probability, and construct a scalar reward $r := (-\boldsymbol{\lambda}^T \boldsymbol{c}_t)$, then any policy that maximizes the expected $r$ is a linear minimizer of (12). Therefore any RL algorithm that best suits the underlying problems can be used to construct a RL oracle.

Certifying constraint satisfaction amounts to evaluate the measurement vector of the current policy. This is handy in online settings, where simulations can be used to evaluate the measurement vector of the policy directly. Otherwise, in batch settings, various off-policy evaluation methods, such as importance sampling (Precup, 2000; Precup et al., 2001) or doubly robust (Jiang & Li, 2016; Dudík et al., 2011), can be used to evaluate the policy.

$$\texttt{RL\_oracle}_\texttt{c}(\boldsymbol{\lambda}) := \boldsymbol{c}(\underset{\pi \in \Pi}{\arg\min} \, \boldsymbol{\lambda}^T \boldsymbol{c}(\pi)) = \underset{\boldsymbol{c}(\pi), \pi \in \Pi}{\arg\min} \, \boldsymbol{\lambda}^T \boldsymbol{c}(\pi). \quad (14)$$

To simplify notation, we assume a RL_Oracle returns a policy as well as its measurement vector

$$\texttt{RL\_Oracle}(\boldsymbol{\lambda}) := \pi, \boldsymbol{c}(\pi) = \texttt{RL\_oracle}_\texttt{p}(\boldsymbol{\lambda}), \texttt{RL\_oracle}_\texttt{c}(\boldsymbol{\lambda}) \quad (15)$$

**Finding Extended Affine Minimizer** The process $\texttt{AffineMinimizer}(S, \boldsymbol{x})$ returns the $(\boldsymbol{y}, \boldsymbol{\alpha})$ the affine minimizer of $S$ with respect to $\boldsymbol{x}$ where $\boldsymbol{y}$ is the affine minimizer and $\boldsymbol{\alpha} := \{\alpha_{\boldsymbol{s}} | \forall \boldsymbol{s} \in \mathcal{S}_c\}$ is the set of coefficient expressing $\boldsymbol{y}$ as an affine combination of points in $S$, that is $\boldsymbol{y} = \sum_{\boldsymbol{s} \in \mathcal{S}_c} \alpha_{\boldsymbol{s}} \boldsymbol{s}$, where $\alpha_{\boldsymbol{s}}$ is the weight associated with $\boldsymbol{s}$. The process $\texttt{AffineMinimizer}(S, \boldsymbol{x})$ can be straightforwardly implemented using linear algebra. Wolfe (1976) also provides a more efficient implementation that uses a triangular array representation of the active set.

## 4.4 CONVERGENCE AND SPARSITY

In this section, we analyze the convergence and complexity of the proposed C2RL method (Algo. 1). We first show that approximation error of C2RL strictly decreases between any two major cycle steps

and it converges in $O(1/t)$ rate. Then we show our method ensures convergence of arbitrary RL algorithm, including those searching for deterministic policies. Moreover, concerning the memory complexity, we show that maintaining an active policy set of $m+1$ is worst case optimal to ensure the convergence of arbitrary RL algorithm. Therefore, the proposed C2RL indeed achieves the optimal sparsity for the found policy, making it favorable for large-scale usage.

The main difference between the convergence analysis of C2RL and Wolfe's method is the addition of the projection step. Intuitively, at each major step, if we are making a significant progress toward the projected point, then the distance to the convex set is decreased by at least the same amount.

**Time Complexity.** In our analysis, we consider the approximation error as defined in (5). We use superscript $t$ to denote the variable in $t$-th major cycle before executing any minor cycle. To simplify notions, we let $x^t := c(\mu^t)$ and $s^t := c(\pi^t)$. When discussing one step with $t$ fixed, let $y^i$ denote the affine minimizer found in $i$-th minor cycle (line 6 of Algo. 1). We first show that the C2RL method strictly reduces approximation error between two calls of the RL oracle.

**Theorem 4.1** (Approximation Error Strictly Decreases). *For any non-terminal step $t$, we have* $\texttt{err}(\mu^{t+1}) < \texttt{err}(\mu^t)$. *That is, the measurement vector of $\mu^t$ found by the C2RL method gets strictly closer to the convex set $\Omega$ after major cycle step.*

The proof is provided in Appendix B. The idea is to consider the distance between $x^t$ and $\omega^t$. When the major cycle has no minor cycle, the non-terminal condition and the affine minimizer property implies $\texttt{dist}^2(x^{t+1}, \omega^t) < \texttt{dist}^2(x^t, \omega^t)$. Otherwise we show that the first minor cycle strictly reduces the $\texttt{dist}^2(x^t, \omega^t)$ by moving along the segment joining $x$ and $y$, and the subsequent minor cycle cannot increase it. Since $\omega^t \in \Omega$, we conclude $\texttt{err}(x^{t+1}) \leq \texttt{dist}^2(x^{t+1}, \omega^t) < \texttt{dist}^2(x^t, \omega^t) = \texttt{err}(x^t)$, and the approximation error strictly decreases.

Given the approximation error strictly decreases, Wolfe's method for minimum norm point can be shown to terminate finitely (Wolfe, 1976). However, this finitely terminating property does not hold for our algorithm. Since a changed $\omega^t$ may yield a lower distance to the same active set $\mathcal{S}_c^t$, the active set may stay unchanged across major cycles (see Figure 2 Middle for an example). Therefore we establish the convergence of the C2RL method by the following theorem.

**Theorem 4.2** (Convergence in Approximation Error). *For $t \geq 1$, the mixed policy $\mu^t$ found by the C2RL method satisfies*

$$\texttt{err}(\mu^t) \leq 16Q^2/(t+2), \tag{16}$$

*where $Q := \max_{\mu \in \Delta(\mathcal{U})} ||c(\mu)||$ is the maximum norm of a measurement vector.*

The proof is provided in Appendix C, which relies on the following two lemmas. We briefly discuss the main idea here. Define major cycle steps with at most one minor cycle as "non-drop step" and major cycle steps with more than one minor cycles as "drop steps". We show that in each non-drop step, Algorithm 1 is guaranteed to make enough progress in the following lemma.

**Lemma 4.3.** *For a non-drop step in C2RL method, we have $\texttt{err}(\mu^t) - \texttt{err}(\mu^{t+1}) \geq \texttt{err}^2(\mu^t)/8Q^2$.*

Though this does not hold for drop steps, we can bound the frequency of drop steps by the following.

**Lemma 4.4.** *After $t$ major cycle steps of C2RL method, the number of drop steps is less than $t/2$.*

Since the approximation error strictly decreases (Thm. 4.1), and in more than half of the major cycles steps, the C2RL method makes significantly progress. The Thm. (4.2) can then be proved using an induction argument (Appendix C).

**Convergence with Arbitrary RL Algo.** The convergence of the C2RL method when used with RL algorithms that search for deterministic policies, such as DQN, DDPG and variants, is indeed straightforward. In (8), though each time the oracle yields a vertex, the FW-type algorithms indeed optimize over the polytope formed by these vertices. Then since citetaltman1999constrained shows that any $c(\cdot)$ achievable can be achieved by some mixed deterministic policies, we conclude that if the underlying problem is feasible, then our C2RL method is able to find a feasible policy.

**Memory Complexity** We then discuss the sparsity of mixed policy for constrained RL problem. We give a constructive proof in Appendix D to show that to ensure convergence for RL algorithms that search for deterministic policies, storing $m+1$ policies is required in the worst case.

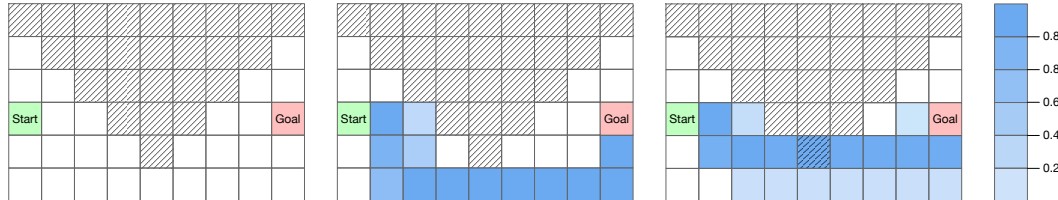

Figure 1: *Left*: The *Risky Mars Rover* environment. The agent is required to navigate from the starting point to reach the goal point without staying long (0.5 steps in expectation) in the risky area (cross-hatching region). *Middle, Right*. Example of an optimal mixed policy found by C2RL in a single run. After 10k samples, C2RL finds a mixed policy that randomizes among two policies with weight $0.49$ and $0.51$. The visitation probabilities of the two policies are plotted.

**Theorem 4.5** (Memory Complexity Bound). *For an constrained RL problem with $m$-dimensional measurement vector, in the worst case, a mixed policy needs to randomize among $m + 1$ individual policies to ensure convergence of RL oracles that search for deterministic policies.*

Since the minor cycles in the C2RL method eliminate policies with affinely dependent measurement vectors, after the termination of minor cycles, the size of the active set is at most $m + 1$. That is, the policy found by the C2RL method requires randomization among no more than $m + 1$ individual policies. Therefore the proposed C2RL indeed achieves the optimal sparsity in the worst case, making it favorable for large-scale usage.

**Corollary 4.5.1.** *The C2RL method that randomizes among at most $m + 1$ policies is worst-case optimal to ensure convergence of any RL oracle.*

## 5 EXPERIMENTS

We evaluate the performance of C2RL in a grid-world navigation task (Fig. 1), and demonstrate its ability to efficiently find sparse policy. In this *Risky Mars Rover* environment, the agent is required to navigate from the starting point to the goal point, by moving to one of the four neighborhood cells at each step. The episodes terminate when the goal point is reached or after $300$ steps. To enforce robustness, we add a risky area to indicate the dangerous states. The agent receives a measurement vector to indicate the steps it takes ($0.1$ for every step), and whether it stays in the risky area ($0.1$ for every risky step, and $0$ otherwise), with discount factor $\gamma = 0.99$. We constrain the agent to reach the goal point with expected cumulative steps measure within $1.1$ and the expected cumulative risky steps within $0.05$. Note that by design, the shortest path from the starting point to the goal point does not satisfy the constraint. This is common in practice, as robustness typically evolves trade-off between the reward and the constraint satisfaction.

The proposed C2RL method is compared with approachability-based policy optimization (ApproPO) (Miryoosefi et al., 2019) and with reward constrained policy optimization (RCPO) (Tessler et al., 2018). ApproPO solves the same convex constrained RL problem by using an RL oracle to play against a no-regret online learner (Hazan et al., 2008; Zinkevich, 2003). Since ApproPO and C2RL both use a RL oracle, ApproPO is a natural baseline to be compared with our method. Besides, we also compare with RCPO, which takes a Lagrangian approach to incorporate the constraints as a penalty signal into the reward. Using an advantage actor critic (A2C) Mnih et al. (2016), RCPO has been shown to converge to a fixed point. For a fair comparison, C2RL and ApproPO uses an A2C agent as the RL oracle, with the same hyperparameter as used in RCPO. The approximation errors are compared after training for the same number of samples.

Note that the C2RL method does not introduce any extra hyper-parameter. For ApproPO and RCPO, they require extra hyper-parameter for the initialization and learning rate of a variable equivalent to our $\lambda$ in the outer loop. This is because our approach does not rely on the online learning framework, and therefore there is no need to tune the initialization and learning rate for our $\lambda$ and ease the usage.

We first showcase the consequences of our theoretical results using an *optimal RL oracle*. For any $x \in \mathbb{R}^m$, an optimal policy can be easily found via Dijkstra's algorithm. If multiple optimal paths exist, one is randomly picked to form a deterministic policy.

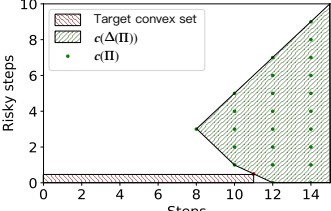 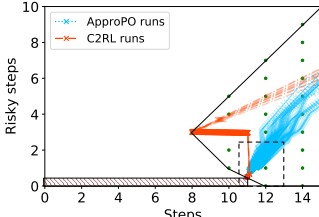 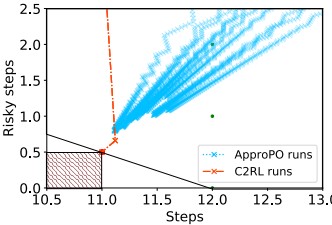

Figure 2: *Left*: Visualization of the distance minimization problem (7) in $\mathbb{R}^2$, where the number of steps and the number of steps in risky zone are measured. The green hatched region is the polytope formed by values achievable by mixed deterministic policies $c(\Delta(\Pi))$, and the red hatched region is the target set. *Middle*: Using an *optimal RL oracle*, 10 paths are sampled to showcase the convergence property of C2RL and ApproPO, where each cross on the dashed line corresponds to a call to the oracle. *Right*: If we zoom in, ApproPO suffers from the zig-zagging problem.

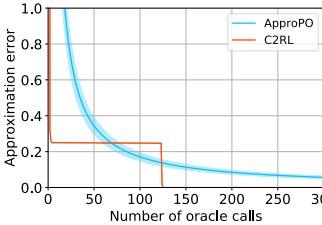 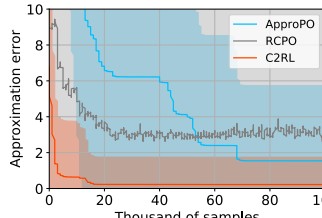 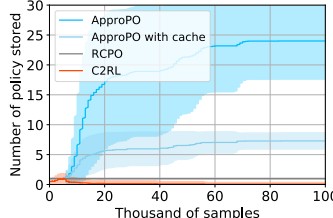

Figure 3: *Left*: Time complexity measured by number of calling an optimal RL oracle. *Middle, Right*: Using A2C to approximate an RL oracle, time complexity measured by thousands of samples and memory complexity measured by the number of policies stored are compared.

Using this as an optimal RL oracle, the convergence property of C2RL and ApproPo are compared. Figure 2 Middle shows the value of policies $c(\mu^t)$ found after each call to the oracle. In Figure 2 Right, when approaching the boundary of the feasible set, the iterations of approachability-based methods start to zigzag. Since C2RL contains a minor cycle to re-optimize the weights among the active set, C2RL progresses quickly to reach the exact optimal solution. In Figure 3 Left, the approximation error is shown for 300 calls of the optimal RL oracle.

We then compare C2RL, ApproPO and RCPO using the same A2C agent (details of the model structures and hyper-parameters are provided in Appendix E). We run each algorithm for 50 times, and each run for a maximum of 100 thousands of samples. The mean and standard deviation of the results are presented in Figure 3. The original paper of ApproPO suggests using a cache to save memory, and the memory requirement of this variant is also presented. Figure 3 demonstrates that C2RL converges to an optimal policy faster than previous methods, and a sparse combination of individual policies is maintained throughout the iteration process.

## 6 CONCLUSION

In this paper, we introduce C2RL, an algorithm to solve RL problems under orthant or convex constraints. Our method reduces the constrained RL problem to a distance minimization problem, and a novel variant of Frank-Wolfe type algorithm is proposed to solve this. Our method comes with rigorous theoretical guarantees and does not introduce any extra hyper-parameter. To find an $\epsilon$-approximation solution, C2RL takes $O(1/\epsilon)$ calls of any RL oracle and ensures convergence to work with arbitrary RL algorithm. Moreover, C2RL strictly reduces the approximation error between consecutive calls of RL oracle, and for $m$-dimensional constraints, the memory requirement is reduced from storing infinitely many policies ($O(1/\epsilon)$) to storing at most constantly many ($m+1$) polices. We further show that the constant is worst-case optimal to ensure the convergence for RL algorithms that search for deterministic policies. Experimentally, we demonstrate that the proposed C2RL method finds sparse solution efficiently, and outperforms previous methods.

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

## A   More on Frank-Wolfe-type Algorithms

### A.1   Standard Frank-Wolfe Algorithm

---

**Algorithm 2** Frank-Wolfe algorithm (Frank et al., 1956)

---

**Input:** obj. $f : \mathcal{Y} \mapsto \mathbb{R}$, oracle $O(\cdot)$, init. $\boldsymbol{x}_0 \in \mathcal{Y}$

1: **for** t=1, 2, 3 ..., T **do**
2:     $\boldsymbol{s} \leftarrow \texttt{Oracle}(\nabla f(\boldsymbol{x}_{t-1})) = \arg\min_{\boldsymbol{s} \in \{\boldsymbol{s}_1,...,\boldsymbol{s}_n\}} \boldsymbol{s}^T \nabla f(\boldsymbol{x}_{t-1})$
3:     $\boldsymbol{x}_t \leftarrow (1 - \eta_t)\boldsymbol{x}_{t-1} + \eta_t \boldsymbol{s}$ , for $\eta_t := \frac{2}{t+2}$
4: **end for**
5: **return** $\boldsymbol{x}_T$

---

For a convex function $f : \mathcal{X} \mapsto \mathbb{R}$ the Frank-Wolfe algorithm (FW) solves the constrained optimization problem over a compact and convex set $\mathcal{X}$. The standard FW is known to have a sublinear convergence rate, and various methods are proposed to improve the performance. For example, when the underlying feasible set is a polytope, and the objective function is strongly convex, multiple variants, such as away-step FW (Wolfe, 1970; Jaggi, 2013), pairwise FW (Mitchell et al., 1974), and Wolfe's method (Wolfe, 1976) are shown to enjoy linear convergence rate. Linear convergence under other conditions is also studied (Beck & Shtern, 2017; Garber & Hazan, 2013a;b).

### A.2   Wolfe's Method for Minimum Norm Point

---

**Algorithm 3** Wolfe's Method for Minimum Norm Point

---

**Initialize** $\boldsymbol{x} \in \mathcal{P}$, active set $\mathcal{S} = [\boldsymbol{x}]$ and weight $\boldsymbol{\lambda} = [1]$.
**Output:** $\boldsymbol{x} \in \mathcal{P}$ that has the minimum Euclidean norm.

1: **while** true **do**                                                                        // Major cycle
2:     $\boldsymbol{s} \leftarrow \texttt{Oracle}(\boldsymbol{x})$                                // Potential improving point
3:     **if** $\|\boldsymbol{x}\|^2 \leq \boldsymbol{x}^T \boldsymbol{s} + \epsilon$ **then** break
4:     $\mathcal{S} \leftarrow \mathcal{S} \cup \{\mathbf{s}\}$
5:     **while** true **do**                                                                    // Minor cycle
6:         $\boldsymbol{y}, \boldsymbol{\alpha} \leftarrow \texttt{AffineMinimizer}(\mathcal{S})$   // $\boldsymbol{y} = \arg\min_{\boldsymbol{s} \in \texttt{aff}(\mathcal{S})} \|\boldsymbol{s}\|_2$
7:         **if** $\alpha_{\boldsymbol{s}} > 0$ for all $\boldsymbol{s}$ **then** break            // $\boldsymbol{y} \in \texttt{conv}(\mathcal{S})$
8:         // If $\boldsymbol{y} \notin \texttt{conv}(\mathcal{S})$, then update $\boldsymbol{y}$ to the intersection of $\texttt{conv}(\mathcal{S})$ and segment joining $\boldsymbol{x}$ and $\boldsymbol{y}$. Then remove points in $\mathcal{S}$ unnecessary for describing $\boldsymbol{y}$.
9:         $\theta \leftarrow \min_{i:\alpha_i \leq 0} \frac{\lambda_i}{\lambda_i - \alpha_i}$     // Recall $\boldsymbol{\lambda}$ satisfies $\boldsymbol{x} = \sum_{\boldsymbol{s} \in \mathcal{S}} \lambda_{\boldsymbol{s}} \boldsymbol{s}$
10:        $\boldsymbol{y} \leftarrow \theta \boldsymbol{y} + (1 - \theta)\boldsymbol{x}, \lambda_i = \theta\alpha_i + (1 - \theta)\lambda_i$
11:        $\mathcal{S} \leftarrow \{\boldsymbol{s}_i | \boldsymbol{s}_i \in \mathcal{S} \text{ and } \lambda_i > 0\}$
12:    **end while**
13:    Update $\boldsymbol{x} = \boldsymbol{y}$ and $\boldsymbol{\lambda} = \boldsymbol{\alpha}$.
14: **end while**
15: **return** $\boldsymbol{x}$

---

Wolfe's method is an iterative algorithm for finding the point with minimum Euclidean norm in a polytope, which is defined as the convex hull of a set of finitely many points.

The Wolfe's method consists of a finite number of major cycles, each of which consists of a finite number of minor cycles. At the start of each major cycle, let $H(\boldsymbol{x}) := \{\boldsymbol{y}^T \boldsymbol{x} = \boldsymbol{x}^{\boldsymbol{x}}\}$ be the hyperplane defined by $\boldsymbol{x}$. If $H(\boldsymbol{x})$ separates the polytope from the origin, then the major cycle is terminated. Otherwise, it invokes an oracle to find any point on the near side of the hyperplane. The point is then added into the active set $\mathcal{S}$, and starts a minor cycle.

In a minor cycle, let $\boldsymbol{y}$ be the point of smallest norm in of the affine hull $\texttt{aff}(\mathcal{S})$. If $\boldsymbol{y}$ is in the relative interior of the convex hull $\texttt{conv}(\mathcal{S})$, then $\boldsymbol{x}$ is updated to $\boldsymbol{y}$ and the minor cycle is terminated. Otherwise, $\boldsymbol{y}$ is updated to the nearest point to $\boldsymbol{y}$ on the line segment $\texttt{conv}(\mathcal{S}) \cap [\boldsymbol{x}, \boldsymbol{y}]$. Thus $\boldsymbol{y}$ is updated to a boundary point of $\texttt{conv}(\mathcal{S})$, and any point that is not on the face of $\texttt{conv}(\mathcal{S})$ in which $\boldsymbol{y}$ lies is deleted. The minor cycles are executed repeatedly until $\mathcal{S}$ becomes a *corral*, that is, a set

whose affine minimizer lies inside its convex hull. Since a set of one point is always a corral, the minor cycles is terminated after a finite number of runs.

## B    Proof of Theorem 4.1

**Theorem 4.1** (Approximation Error Strictly Decreases). *For any non-terminal step $t$, we have* $\texttt{err}(\mu^{t+1}) < \texttt{err}(\mu^t)$. *That is, the measurement vector of $\mu^t$ found by the C2RL method gets strictly closer to the convex set $\Omega$ after major cycle step.*

*Proof.* If the current step is a major cycle with no minor cycle, then $x^{t+1}$ is the affine minimizer of $\texttt{aff}(\mathcal{S} \cup \{s^t\})$ with respect to $\omega^t$. Then the affine minimizer property implies $(s^t - x^{t+1})(x^{t+1} - \omega^t) = 0$. Since iteration does not terminate at step $t$, we have $(x^t - \omega^t)^T(x^t - s^t) > 0$, and therefore $x^{t+1}$ not equal to $x^t$. Then $x^{t+1}$ is the unique affine minimizer implies $f_\Omega(x^{t+1}) = \min_{\omega \in \Omega} ||x^{t+1} - \omega||^2 \le ||x^{t+1} - \omega^t||^2 < ||x^t - \omega^t||^2 = f_\Omega(x^t)$.

Otherwise the current step contains one or more minor cycles. In this case, we show that the first minor cycle strictly reduces the approximation error, and the (possibly) following minor cycles cannot increase it. For the first minor cycle, the affine minimizer $y^0$ of $\texttt{aff}(\mathcal{S} \cup \{s^t\})$ with respect to $\omega^t$ is outside $\texttt{conv}(\mathcal{S} \cup \{s^t\})$. Let $z = \theta y^0 + (1 - \theta)x^t$ be the intersection of $\texttt{conv}(\mathcal{S} \cup \{s^t\})$ and segment joining $x$ and $y$. Let $\mathcal{V}^0 := \mathcal{S}^t$ and $\mathcal{V}^i$ denote the active set after the $i$-th minor cycle. Then since $y^1$ is the affine minimizer of $\mathcal{V}^1$ with respect to $\omega^t$, we have

$$||z - \omega^t|| = ||\theta y^0 + (1 - \theta)x^t - \omega^t|| \le \theta||y^0 - \omega^t|| + (1 - \theta)||x^t - \omega^t|| < ||x^t - \omega^t||, \quad (17)$$

where the second step uses the triangle inequality and the last step follows since the segment $x^t y^0$ intersects the interior of $\texttt{conv}(\mathcal{S} \cup \{s^t\})$, and the distance to $\omega^t$ strictly decreases along this segment. Therefore the point $z$ found by first minor cycle satisfies

$$f_\Omega(z) = \min_{\omega \in \Omega} ||z - \omega||^2 \le ||z - \omega^t||^2 < ||x^t - \omega^t|| = f_\Omega(x^t). \quad (18)$$

Hence $h(y^1) < h(x^t)$, and the first minor cycle strictly decreases the approximation error. By a similar argument, in subsequent minor cycles the approximation error cannot be increased. However, after the first minor cycle, the iterating point may already at the intersection point and the strict inequality in last step of Eq. 17 need to be replaced by non-strict inequality.

Therefore any major cycle either finds an improving point and continue, or enters minor cycles where the first minor cycle finds an improving point, and the subsequent minor cycles does not increase the distance. Adding both side of $f_\Omega(x^{t+1}) < f_\Omega(x^t)$ by $f_\Omega(x^*)$ and we have the approximation error $h(x^{t+1}) < h(x^t)$ strictly decreases.    □

## C    Proof of Theorem 4.2

We first prove the Theorem 4.2, using Lemma 4.3 and Lemma 4.4. Then we present the proof of the lemmas.

**Theorem 4.2** (Convergence in Approximation Error). *For $t \ge 1$, the mixed policy $\mu^t$ found by the C2RL method satisfies*

$$\texttt{err}(\mu^t) \le 16Q^2/(t + 2). \quad (19)$$

*where $Q := \max_{\mu \in \Delta(\mathcal{U})} ||c(\mu)||$ is the maximum norm of a measurement vector.*

*Proof.* Since Lemma 4.4 shows that drop steps are no more than half of total major cycle steps, and Theorem 4.1 guarantees these drop steps reducing the approximation error, we can safely skip these step, and re-index the step numbers to include non-drop steps only using $k$.

For these non-drop steps, we claim that $\text{err}(\mu^k) \leq 8Q^2/(k+1)$. Using Lemma 4.3, we prove the convergence rate using induction. We first bound the error of any $\text{err}(\mu^k)$. For any $k \geq 1$

$$\text{err}(\mu^k) = \text{dist}^2(\boldsymbol{c}(\mu^k), \Omega) - \text{dist}^2(\boldsymbol{c}(\mu^*), \Omega) \tag{20}$$

$$= 1/2\|\boldsymbol{c}(\mu^k) - \text{Proj}_\Omega(\boldsymbol{c}(\mu^k))\|^2 - 1/2\|\boldsymbol{c}(\mu^*) - \text{Proj}_\Omega(\boldsymbol{c}(\mu^*))\|^2 \tag{21}$$

$$\leq 1/2(\|\boldsymbol{c}(\mu^k)\|^2 + \|\text{Proj}_\Omega(\boldsymbol{c}(\mu^k))\|^2 - \|\boldsymbol{c}(\mu^*)\|^2 - \|\text{Proj}_\Omega(\boldsymbol{c}(\mu^*))\|^2) \tag{22}$$

$$\leq \|\boldsymbol{c}(\mu^k)\|^2 - \|\boldsymbol{c}(\mu^*)\|^2 \tag{23}$$

$$\leq \|\boldsymbol{c}(\mu^k)\|^2 \tag{24}$$

$$\leq Q^2, \tag{25}$$

where Eq. 21 uses the definition of our squared Euclidean distance function. Eq. 22 follows from triangle inequality, and Eq. 23 is by the contractive property of the Euclidean distance.

When $k = 1$, the Eq. 25 established the based case. Now for $k \geq 1$, assume that $\text{err}(\mu^k) \leq 8Q^2/(k+1)$ for $k \geq 1$, then Lemma 4.3 gives $\text{err}(\mu^{k+1}) \leq \text{err}(\mu^k) - \text{err}^2(\mu^k)/8Q^2$. Since the quadratic function of the right hand side is monotonically increasing on $(-\infty, 4Q^2]$, using the inductive hypothesis

$$\text{err}(\mu^{k+1}) \leq \text{err}(\mu^k) - \text{err}^2(\mu^k)/8Q^2 \leq 8Q^2/(k+1) - 8Q^2/(k+1)^2 \leq Q^2/(k+2) \tag{26}$$

Since for $t$ steps of major cycle steps, the number of non-drop steps $k > t/2$, we conclude that $\text{err}(\mu^t) \leq 16Q^2/(t+2)$.

$\square$

Then we prove the lemmas.

**Lemma 4.3.** *For a non-drop step, we have* $\text{err}(\mu^t) - \text{err}(\mu^{t+1}) \geq \text{err}^2(\mu^t)/8Q^2$.

*Proof.* The non-drop step contains either no minor cycle or one minor cycle. We first consider the no minor cycle case.

If a major cycle contains no minor cycle, then $\boldsymbol{x}^{t+1}$ is the affine minimizer of the $\mathcal{S} \cup \{\boldsymbol{s}^t\}$.

$$\text{err}(\mu^t) - \text{err}(\mu^{t+1}) = \text{dist}^2(\boldsymbol{x}^t, \Omega) - \text{dist}^2(\boldsymbol{x}^{t+1}, \Omega) \tag{27}$$

$$= 1/2(\|\boldsymbol{x}^t - \boldsymbol{\omega}^t\|^2 - \min_{\boldsymbol{\omega} \in \Omega} \|\boldsymbol{x}^{t+1} - \boldsymbol{\omega}\|^2) \tag{28}$$

$$\geq 1/2(\|\boldsymbol{x}^t - \boldsymbol{\omega}^t\|^2 - \|\boldsymbol{x}^{t+1} - \boldsymbol{\omega}^t\|^2) \tag{29}$$

$$= 1/2(\|\boldsymbol{x}^t - \boldsymbol{\omega}^t\|^2 + \|\boldsymbol{x}^{t+1} - \boldsymbol{\omega}^t\|^2 - 2\|\boldsymbol{x}^{t+1} - \boldsymbol{\omega}^t\|^2) \tag{30}$$

$$= 1/2(\|\boldsymbol{x}^t - \boldsymbol{\omega}^t\|^2 + \|\boldsymbol{x}^{t+1} - \boldsymbol{\omega}^t\|^2 - 2(\boldsymbol{x}^t - \boldsymbol{\omega}^t)^T(\boldsymbol{x}^{t+1} - \boldsymbol{\omega}^t)) \tag{31}$$

$$= 1/2(\|\boldsymbol{x}^t - \boldsymbol{x}^{t+1}\|^2), \tag{32}$$

where the equation (31) follows from the affine minimizer property Eq. (11). For $\|\boldsymbol{x}^t - \boldsymbol{x}^{t+1}\|$ in the last equation, and $\forall \boldsymbol{q} \in \text{aff}(\mathcal{S} \cup \{\boldsymbol{s}^t\})$, we have

$$\|\boldsymbol{x}^t - \boldsymbol{x}^{t+1}\| \geq \|\boldsymbol{x}^t - \boldsymbol{x}^{t+1}\|\frac{\|\boldsymbol{x}^t\| + \|\boldsymbol{q}\|}{2Q} \qquad (\text{Definition of } Q) \tag{33}$$

$$\geq \|\boldsymbol{x}^t - \boldsymbol{x}^{t+1}\|\frac{\|\boldsymbol{x}^t - \boldsymbol{q}\|}{2Q} \qquad (\text{Triangle inequality}) \tag{34}$$

$$\geq \frac{1}{2Q}(\boldsymbol{x}^t - \boldsymbol{x}^{t+1})(\boldsymbol{x}^t - \boldsymbol{q}) \qquad (\text{Cauchy-Schwarz inequality}) \tag{35}$$

$$= \frac{1}{2Q}(\boldsymbol{x}^t - \boldsymbol{\omega}^t)(\boldsymbol{x}^t - \boldsymbol{q}) \qquad (\text{Affine minimizer property}). \tag{36}$$

Then it suffices to show that $(\boldsymbol{x}^t - \boldsymbol{\omega}^t)(\boldsymbol{x}^t - \boldsymbol{q}) \geq \text{err}(\mu^t)$.

Since $\Omega$ is a convex set, the squared Euclidean distance function $\mathtt{dist}^2(\boldsymbol{x}, \Omega)$ is convex for $\boldsymbol{x}$, which implies

$$\mathtt{dist}^2(\boldsymbol{x}^t, \Omega) + (\boldsymbol{q} - \boldsymbol{x}^t)\nabla\mathtt{dist}^2(\boldsymbol{x}^t, \Omega) \le \mathtt{dist}^2(\boldsymbol{q}, \Omega). \tag{37}$$

Putting in $\nabla\mathtt{dist}^2(\boldsymbol{x}^t, \Omega) = (\boldsymbol{x}^t - \mathtt{Proj}_\Omega(\boldsymbol{x}^t)) = (\boldsymbol{x}^t - \boldsymbol{\omega}^t)$, we get $(\boldsymbol{x}^t - \boldsymbol{\omega}^t)(\boldsymbol{x}^t - \boldsymbol{q}) \ge \mathtt{err}(\mu^t)$, which together with Eq. 32 and Eq. 36 concludes that for non-drop step with no minor cycles, we have $\mathtt{err}(\mu^t) - \mathtt{err}(\mu^{t+1}) \ge \mathtt{err}^2(\mu^t)/8Q^2$.

For non-drop step with one minor cycle, we use the Theorem 6 of (Chakrabarty et al., 2014). By a linear translation of adding all points with $-\omega^t$, it gives

$$||\boldsymbol{x}^t - \boldsymbol{\omega}^t||^2 - ||\boldsymbol{x}^{t+1} - \boldsymbol{\omega}^t||^2 \ge ((\boldsymbol{x}^t - \boldsymbol{\omega}^t)(\boldsymbol{x}^t - \boldsymbol{q}))^2/8Q^2. \tag{38}$$

Then applying the same argument as Eq. 37, and we finished our proof.

$\square$

**Lemma 4.4.** *After $t$ major cycle steps of C2RL method, the number of drop steps is less than $t/2$.*

*Proof.* Recall that at the termination of a minor cycle, the size of the active set $|\mathcal{S}_c| \in [1, m]$. Since in each major cycle steps, the size of active set $\mathcal{S}_t$ increases by one, and each drop step reduces the size of $\mathcal{S}_t$ by at least one, the number of drop steps is always less than half of total number of the major cycle steps. $\square$

# D    PROOF OF THEOREM 4.5

**Theorem 4.5** (Memory Complexity Bound). *For an constrained RL problem with $m$-dimensional measurement vector, in the worst case, a mixed policy needs to randomize among $m + 1$ individual policies to ensure convergence of RL oracles that search for deterministic policies.*

*Proof.* We give a constructive proof. Consider a $m$-dimensional vector-valued MDP with a single state, $m + 1$ actions, and $\boldsymbol{c}(a_i) := \boldsymbol{e}_i$ is the unit vector of $i$-th dimension for $i \in [1, m]$, and $\boldsymbol{c}(a_{m+1}) := \boldsymbol{0}$, and the episode terminates after 1 steps. The constrained RL problem is to find a policy whose measurement vector lies in the convex set of a single point $\{\boldsymbol{1}/2m\}$. By linear programming, it is clear that the only feasible mixed deterministic policy is to select $a_{m+1}$ with $1/2$ probability, and the rest $m$ actions with $1/2m$ probability; i.e. the unique feasible policy to this problem has an active set containing $m + 1$ deterministic policies. Therefore any method randomize among less than $m + 1$ individual policies does not ensure convergence when used with RL algorithms searching for deterministic policies. $\square$

# E    ADDITIONAL EXPERIMENT DETAILS

All the methods use the same A2C agent. The input is the one-hot encoded current position index. The A2C is the standard fully connected multi-layer perceptron with ReLU activation function. The actor and critic share the internal representation and have their only final layer. Both actor and critic networks use Adam optimizer with learning rate set to $1e^{-2}$. The network is as follows

|  | Actor | Critic |
|---|---|---|
| Input layer | One-hot encoded state index (dim=54) | |
| Hidden layer | Linear(in=54, out=128, act="relu") | |
| Output layer | Linear(in=128, out=4, act="relu") | Linear(in=128, out=1, act="relu") |
| Output name | Action score | State value |

For ApproPO, the constant $\kappa$ for projection convex set to convex cone is set to be 20. The $\boldsymbol{\theta}$ is initialized to 0. Following the original paper.

For RCPO, the learning rate of its $\boldsymbol{\lambda}$ is set to $2.5e^{-5}$, and its $\boldsymbol{\lambda}$ is initialized to 0 and updated by online gradient descent with learning rate set to 1, as used by the original paper.

The proposed C2RL introduces no extra hyper-parameters, and has nothing to report.

