# OpenReview forum: "A Reduction Approach to Constrained Reinforcement Learning"
_ICLR.cc/2021/Conference — Reject_

### Official Review · AnonReviewer4 · 2020-10-16
**Marginally above the acceptance threshold**

**Rating:** 6
**Confidence:** 3

**Review:**

1. Summarize what the paper claims to do/contribute. Be positive and generous.

In this paper, the authors consider a class of constrained MDP problem. In the considered problem, instead of getting a scalar reward, in each step, the MDP returns a vector “reward” which is termed as a measurement vector. Then the problem requires finding a (mixed) policy such that the expected measurement vector belongs to a convex set. This problem was first considered in (Miryoosefi et al. , 2019) and the authors of this paper propose a new algorithm and claim an improvement over the sparsity in the mixed policy. The main idea of the new algorithm is to solve the problem as convex minimization of the squared distance to the target set. A standard Frankwolfe-type algorithm is proposed to solve the problem, convergence and complexity analysis are also provided.

2. Clearly state your decision (accept or reject) with one or two key reasons for this choice.
This paper is marginally above the acceptance threshold.

3. Provide supporting arguments for the reasons for the decision.

(i). (Weakness) A significant flaw is the WRONG claim of improvement over the previous result (Miryoosefi et al. , 2019). In this paper, the authors yield dist(c(\mu),\Omega) \leq 1/T. While (Miryoosefi et al. , 2019) proves dist(c(\mu),\Omega) \leq 1/sqrt{T}. Thus the authors claim an improved O(1/\epsilon) complexity over the O(1/\epsilon^2) complexity of the compared paper. However, this is a wrong argument. If the authors pay attention to the definition of (Miryoosefi et al. , 2019), they should find that the “dist” denotes the standard notation of Euclidean distance. However, the “dist” function in this paper is the SQUARED Euclidean distance. They are defined differently. Therefore, if we view the two results under the same optimality measure, these two complexity results are the same. No improvement is made in this paper in terms of complexity.

(ii). (strength) Though in (i), we find there is no improvement in terms of complexity and hence the sparsity of the mixed strategy. The reason why I still think it is marginally above the threshold is that, compared to the existing approach in (Miryoosefi et al. , 2019), the Frank-Wolfe type algorithm is way more natural and robust.

(iii). (strength) Due to the desirable structure of the Frank-Wolfe method, the authors are able to constantly eliminate the affinely dependent historical policies/measurement vectors. Thus limiting the storage memory to only m+1, where m is the dimension of the measurement vector.

4. Provide additional feedback with the aim to improve the paper. Make it clear that these points are here to help, and not necessarily part of your decision assessment.

(i). The author should change their claim of the improvement over (Miryoosefi et al. , 2019), such improvement is not achieved in this paper.

(ii). In terms of notation, the definition of the “dist” function should be changed. Traditionally, “dist” only denotes the distance instead of the squared distance. The authors should use “dist^2(c,x)” or simply change to some other notation, otherwise, it will cause confusion. (This notation confusion is also possibly the reason why the authors wrongly claim the improvement over (Miryoosefi et al. , 2019))

(iii). It might worth more discussion about what Algorithm 1 is doing. For example, for line 3, briefly comment that $x-\omega$ is the gradient of the objective function will make the understanding of the algorithm significantly simpler. Similarly, the authors can make more explanation about the line 6-13 of Algorithm 1, which might cause confusion without explanation.

5. Ask questions you would like answered by the authors that help you clarify your understanding of the paper and provides the additional evidence you need to make be confident in your assessment.

Consider finding a policy \pi whose measurement vector c(\pi)\in\Omega. For simplicity suppose there are only 2 policies. In this paper, the proposed solution is to find p s.t.
p*c(\pi) + (1-p)*c(\pi’)\in\Omega. However, this corresponds to the situation where before doing anything, first toss a coin to decide whether \pi or \pi’ is used. Then use that policy for all future plays. However, this situation is weird in the sense that none of the \pi or \pi’ is feasible, and the variance can be very large. My question is, is it possible to find the convex combination s.t. c(p*\pi + (1-p)*\pi’)\in\Omega? (It is obvious that c(p*\pi + (1-p)*\pi’) \neq p*c(\pi) + (1-p)*c(\pi’)). I think such a policy will be stabler.

---

> ### Author Response · Authors · 2020-11-19
> **Thank you for your helpful comment**
>
>
> We thank the reviewer for his/her constructive comments and feedback.
>
> 1. Time complexity and dist function notation.
> You are right, our reference to time complexity in Miryoosefi’s method was erroneous, and we have fixed it as well as the dist function notation (to dist^2). However, this does not affect the main results of our method. The main contribution of our method is that, by proposing a new reduction approach and a new variant FW type algorithm, we can find feasible policy using constantly many individual policies, without the need to introduce any extra hyper-parameter.
>
> 2. More discussion about Algo 1.
> Thank you for the feedback. We have modified Sec 4.2 and 4.3 with more discussions and explanations. The main difference between our method and Wolfe’s method is the addition of a projection step in the major cycle. In each major cycle step, we minimize the distance toward the projected point. For line 6-13, we use the same minor cycle as Wolfe’s original method, and details are provided in Appendix A.2.
>
> 2. Stabler way to combine two policies.
> In tabular case, convex combination of two policies can be straightforward. However, it is non-trivial in high-dimension, especially when neural networks are used. For example, to come up with proper weights for choosing each actions requires accurately estimating the state visitation frequency, which is non-trivial for off-policy methods.
> If the goal is to find a stabler policy, in fact there is a simpler way. Currently, we show that when deterministic policies are used, it requires randomization among at most $m+1$ policies. In fact, when considering stationary policies, a stronger conclusion can be obtained. Thm. 3.8 of Altman (1999) proves that there exists a stationary policy that requires randomization in at most $m$ states. When considering mixed deterministic policies, this implies that it is possible to find a mixed deterministic policy that consists at most $m+1$ deterministic policies, which behave identically except in at most $m$ states. This can be implemented by adding constraints on the policies in the active set to find a stabler mixed policy, and we leave it for further study.
>
> Eitan Altman.Constrained Markov decision processes, volume 7. CRC Press, 1999.

---

### Official Review · AnonReviewer2 · 2020-10-19
**A good paper with solid theoretical improvement**

**Rating:** 7
**Confidence:** 2

**Review:**

This paper presents a reduction approach to tackle the optimization problem of constrained RL. They propose a Frank-Wolfe type algorithm for the task, which avoids many shortcomings of previous methods, such as the memory complexity. They prove that their algorithm can find an $\epsilon$-approximate solution with $O(1/\epsilon)$ invocation. They also show the power of their algorithm with experiments in a grid-world navigation task, though the tasks looks relatively simple.

pros:
- The application of Frank-Wolfe algorithm to constrained RL problem is novel. The method is basically different from the that of Miryoosefi et al. (2019).The improvement is mainly due to the algorithm design.

- The theoretical improvement is solid. The paper tackles the memory requirement  issue in the previous literature, and only requires constant memory complexity. Further, the number of RL oracle invocation is also reduced from $O(1/\epsilon^2)$ to $O(1/\epsilon)$.

- The paper is well-written. Though I only sketched the proof in the appendix, the algorithm and the analysis in the main part is reasonable and sound.

comments:
- The algorithm requires a policy set $\mathcal{U}$ and finds a mixed policy $\mu \in \Delta(\mathcal{U})$ to satisfy the constraints. How to get a policy set with a feasible solution? Is $\mathcal{U}$ predefined? For an MDP with $S$ states and $A$ actions, the possible deterministic policy can be $A^S$. Trivially setting $\mathcal{U}$ as a set with all possible policies may lead to exponential computational and memory complexity.

- Constrained RL problem can be formulated from the standard dual LP form of RL problem, in which the policy $\pi$ can be fully represented as the density over state-action $d(s,a)$ (See e.g. [1]). Is it possible to solve constrained RL problem under this formulation? What is the advantage of using mixed policies over fixed policy set $\mathcal{U}$ compared with this formulation?

typos:
- line 3 of Algorithm 2: $(1-\eta_t w_{t-1})$ -> $(1-\eta_t) w_{t-1}$

[1] Constrained episodic reinforcement learning in concave-convex and knapsack settings

---

> ### Author Response · Authors · 2020-11-19
> **Thank you for your helpful comment**
>
>
> 1. How to get a policy set with a feasible solution?
> For a feasible constrained RL problem, there always exists feasible solution in stationary policy and mixed deterministic policies (Theorem 4.2 of Altman (1999)). Therefore, to utilize RL algorithms that search for deterministic policies, such as DQN, DDPG and variants, in constrained RL problems, we have to store multiple such policies, and choose a mixed strategy according to some probability. However, for stationary policies that choose actions with certain probability at each state, there always exists feasible solutions.
>
> 2. How to choose policy set?
> You don’t have to. The discussion of policy set is to ensure the convergence of our algorithm when various RL algorithms are used to construct a RL oracle. Though RL algorithms that search for deterministic policy (such as DQN/DDPG) may not find a feasible policy for the constrained problem (because our method actually optimizes over the convex polytope formed by these deterministic policies), our method ensures convergence when such RL algorithms are used to construct an RL oracle. Users can simply pick up their favorable RL algorithm for the target problem, and use it to construct a RL oracle, then they are ready to solve a constrained RL problem using the proposed C2RL method. (We added discussion for this right after Lemma 4.4).
>
> 3. Solving constrained RL with policies represented as the density over state-action d(s,a).
> Such approach is similar to the first group of methods in our related work section, and is tied to specific families of RL algorithms. Policies representable by density over state-action is in fact all stationary policies. To see this, for any state $s$, consider the policy $\pi(a|s) := d(s,a) / \sum_{s} d(s,a)$, it is clear that there is a one-to-one correspondence between all stationary policies and all possible densities over state-action. In a tabular case, it is possible to store all state-action density. In high-dimension, this corresponds to using a policy network, such as the A2C agent used in RCPO. The main limitation of these methods is that they cannot ensures convergence when used with RL algorithms that find deterministic policies (value function methods in tabular case, DQN/DDPG etc. in high dimensional case), and are tied to policy optimization methods. In the past few year, we have witnessed the booming development of RL algorithms; therefore, we think a general solution that does not tie to any family of RL algorithms is more favorable.
>
> Thank you for your additional points. We have reflected them in the paper.
>
> Eitan Altman.Constrained Markov decision processes, volume 7. CRC Press, 1999.

---

### Official Review · AnonReviewer3 · 2020-10-24
**Section 4 is hard to parse**

**Rating:** 5
**Confidence:** 2

**Review:**

The authors propose C2RL to solve RL problems under convex constraints. The authors reduce the RL problem under convex constraints to a distance minimization problem and solve the distance minimization problem with a Frank-Wolfe type method (with an RL solver as a sub-routine).
The authors further show that the algorithm converges (in terms of approximation error), and validate some of their theoretical findings with simulations.

# Pros:
I like the fact that the authors have theoretical guarantees for their approximation error.
The reduction to distance minimization problem is also clear.

# Cons:
+ I find it hard to understand Section 4.3 and Section 4.4 even after quite a few passes.
This is the main reason for my score and my confidence level. For Section 4.3, while I understand how Algorithm 1 works, I have no intuitive idea why Algorithm 1 converges. There is a lack of connection to the original Wolfe's algorithm (such as what corresponds to the objection function and to the linear minimization oracle? Why the linear property of the RL-oracle is important? etc). For Section 4.4, the authors just pile-up their results without further remark on the implications.

+ I don't understand the role of the sparse policy here. Does finding a sparse policy makes the problem easier or harder? Why do we want to find a sparse policy?

+ It seems that the optimal $\mu$ is not unique, if there are multiple $\mu$ such that $c(\mu) \in \Omega$. If this is so, the analysis of the Frank-Wolfe type method could be tricky.

+ While I understand the challenges of RL problem under convex constraints, could the author list specifically what are the applications that can be formulated into RL under cvx constraints? Do we have an easy projection operator for these convex constraints? How to choose the policy set in the real world application?

Minor comments:

+ Above equation (7): “is equivalent to minimizing the distance between the polytope and the convex set”. It is misleading to talk about the distance between the two sets. Maybe "find a point in the polytope that is closest to the convex set $\Omega$"?

+ Some comments on the meaning of equation (4) should be helpful for the readers to understand the main flow

---

> ### Author Response · Authors · 2020-11-19
> **Thank you for your helpful comment, we address your concerns in our comment below**
>
>
> We thank the reviewer for his/her constructive comments and feedback.
>
> 1. Intuitively, why Algo. 1 converges?
> The convergence of Algo. 1 can be better understood when derived from the standard FW algorithm. Since our objective function is the half squared distance function to a convex set $\Omega$, its gradient is $x-\omega := x – Proj_{\Omega}(x)$, and RL_Oracle($x-\omega $) is the improving point found by the linear minimization oracle. Then, at each step, by treating the $Proj_{\Omega}(x)$ as the origin, we use the same minor cycle as Wolfe’s method, to reweigh the policies in the active set such that the weighted measurement vector is closest to $Proj_{\Omega}(x)$. Intuitively, since the distance to the convex set is upper bounded by the distance to this projected point, if the distance to the projected point converges, so does the distance to the target convex set. We modified and added more discussions in Sec 4.2 and 4.3 to make the flow clearer. Implications of theoretical analysis are added in Sec 4.4.
>
> 2. Why the linear property of RL-oracle is important?
> The linear minimizer oracle of a FW type algorithm requires finding a measurement vectors $c(\pi)$ that minimizes $\lambda^T c(\pi)$. However, a typical RL algorithm only deals with a scalar reward, instead of a measurement vector $c_t$ after each state transition. The linear property links them and shows that we can construct a RL oracle from any RL algorithm. If we use $r = \lambda^T c_t$ as a scalar reward for each transition, then any RL algorithm that finds an optimal policy that maximizes expected $r$, finds a linear minimizer for $\lambda^T c(\pi)$. Therefore, we can construct a linear minimizer oracle, a RL oracle, using arbitrary RL algorithms. (Modified and discussed in Sec 4.3)
>
> 3. Why sparsity is a major concern in constrained RL?
> Please kindly refer to our response to AnonReviewer1’s comment 1.
>
> 4. Is finding sparse policy harder or easier?
> It is harder. In fact, finding more sparse policy is impossible in previous game-theoretical framework. The convergence of previous approaches (Agarwal, Le and Miryoosefi) relies on the no-regret property of an online learner, making it impossible to eliminate any individual policy and thus cannot achieve sufficient sparsity. Our approach reduces the constrained problem to a distance minimization problem and solve it with FW type algorithms. Getting rid of the online learning framework enables elimination of unnecessary polices, and achieves better sparsity.
>
> 5. Optimal $\mu$  is not unique?
> Yes, optimal $\mu$ may not be unique, and it is common in practice that even optimal $c(\mu)$ is not unique. Since we define the constrained RL problem as a feasibility problem, any policy that satisfies $c(\mu) \in \Omega$ is a feasible policy, and we did not see any tricky part.
>
> 6. Any convex constraints applications?
> For example, exploration suggestions that require visiting all states as evenly as possible, are naturally modelled by constraining the distance between the state visitation frequency and an uniform distribution within a convex set. In fact, the commonly studied constrained RL problem, such as maximizing a reward subject to an inequality constraint, can be reformulated to our problem easily, which involves binary search on the reward, and using our proposed C2RL to find feasible policy. (We added this convex constraint example in Sec 1)
>
> 7. Why not projection method?
> Please kindly refer to our response to AnonReviewer1’s comment 5.
>
> 8. How to choose a policy set?
> You don’t have to. The discussion of policy set is to ensure the convergence of our algorithm when various RL algorithms are used to construct a RL oracle. Though RL algorithms that search for deterministic policy (such as DQN/DDPG) may not find a feasible policy for the constrained problem (because our method actually optimizes over the convex polytope formed by these deterministic policies), our method ensures convergence when such RL algorithms are used to construct an RL oracle. Users can simply pick up their favorable RL algorithm for the target problem, and use it to construct a RL oracle, then they are ready to solve a constrained RL problem using the proposed C2RL method. (We added discussion for this right after Lemma 4.4).
>
> Thank you for your other points. We have incorporated those changes into the paper.

---

### Official Review · AnonReviewer1 · 2020-10-27
**The paper proposes a fast method for solving reinforcement learning problems with constraints. Oracle computational and memory complexity of the proposed algorithm are provided along with experiments on a grid-world navigation task to illustrate the convergence behavior of the proposed algorithm.**

**Rating:** 5
**Confidence:** 4

**Review:**

Summary: The paper proposes a fast method for solving reinforcement learning problems with constraints. Oracle computational and memory complexity of the proposed algorithm are provided along with experiments on a grid-world navigation task to illustrate the convergence behavior of the proposed algorithm.

Strength:

- Instead of using a penalty or a regularization scheme to impose safety, risk or budget constraints, the paper proposes to impose the constraints explicitly, or in other words treat them as hard constraints. Dealing with hard constraints is an important open problem especially in nonconvex problems as in reinforcement learning. The paper uses the reductions approach proposed in Agarwal et al, to transform the policy constrained problem to a mixed policy scheme in which finding a feasible solution is equivalent to finding a distribution over policies. Now, with this reduction, the paper leverages the fact that feasible set is guaranteed to be a polytope (given by the convex hull of all feasible policies), and hence the paper solves the equivalent problem of finding a point that minimizes the distance to the convex polytope.

- The minimum norm point algorithm or Wolfe's method is proposed to solve the distance minimization problem. An advantage of the algorithm is that it can be used in tandem with any off-the-shelf RL algorithm while guaranteeing feasibility unlike the penalty based methods. In addition to the time complexity of the algorithm, they also provide a memory complexity bound which in some sense is inherited from the minimum norm point algorithm. The analysis in the appendix seem to use the techniques from the Chakrabarty et al paper.

Weaknesses:

- There is no discussion or comparison of subproblem complexity with respect to existing methods. I consider this paper to be a theoretical paper, and it is unfortunate that there is no mention to any practical use cases. I believe that this is an important aspect of numerical algorithm that needs to be discussed in papers. For example, it is well known that exact projections do provide strictly feasible solutions and solve a problem that is in theory equivalent. That is, if the feasible set is a polytope, Euclidean projection and minimum norm point are both generic quadratic programming problems, hence it is not clear why the minimum norm point should be preferred in reinforcement learning settings. I believe that sparsity in the intermediate iterates seem like the crucial difference, but the paper discusses this aspect merely in passing. Some specific examples of safety, risk or budget constraints and describing the subproblem complexity for such constraints will make this clear.

- I'm not sure if the experiments are reproducible with the information provided in the paper. First, the experiments seem separate from the rest of the paper and the reader has to go over other papers to get an idea of the overall setup. For example, it is not clear why 300 steps are sufficient or what risky region is. Secondly, there is no discussion of the hyperparameters used in the experiments and algorithm. For example, how was the value of epsilon determined? the step size or learning rate? I think the paper would hugely benefit from ablation studies in more than one task, preferably something that is used in practice.

After response: Thanks for the clarifications. However, conceptually important questions are not yet clarified yet. For example,  the objective itself can be made into a pure square function (and hence strongly convex) in both classical Wolfe's formulation and the proposed. As the authors are pointing out, the main issue is in designing separation (or projection) oracle for the constraints which corresponds to the base polytope in the context of submodular optimization and was the main motivation for Wolfe's algorithm. Moreover, authors mention that main difficulty in using projections is intractability but it is not clear why the linear optimization performed in the proposed algorithm is efficient.

---

> ### Author Response · Authors · 2020-11-19
> **Thank you for your helpful review, clarification provided in the comment**
>
>
> We thank the reviewer for his/her constructive comments and feedback. It seems that our original paper has caused a few misunderstandings. We would like to clarify our thoughts here and have made corresponding revisions in the latest version of the paper.
>
> 1. Why sparsity is a major concern in constrained RL?
> Our research was originally motivated by applying DQN for an online marketing campaign to optimize a long-term user activeness metric subject to a budget constraint. In our case, existing methods (Le’s and Miryoosefi’s) require storing a few hundred Q-networks to make a single decision, where each Q-network costs about 1GB memory. But it is practically infeasible to train, deploy and make inference using so many networks at large-scale. That is why sparsity is a key focus point of our solution. Using our proposed C2RL method, for example, in a 1-dimensional constraint problem like budget, we only need to store at most two Q-networks, which is more favorable for industrial usage.
>
> 2. Any practical use cases?
> In fact, the commonly studied constrained RL problem, such as maximizing a reward subject to an inequality constraint, can be reformulated to our problem easily, which involves binary search on the reward, and using our proposed C2RL to find feasible policy. In addition to orthant constraints, convex constraints are also studied previously. For example, exploration suggestions that require visiting all states as evenly as possible, are naturally modelled by constraining the distance between the state visitation frequency and an uniform distribution within a convex set.
>
> 3. Differences between our reduction approach and Agarwal’s.
> Agarwal’s approach reformulates the constrained problem to a repeated game, and uses online learning to find its equilibrium. In constrained RL literature, Le’s and Miryoosefi’s methods take the same approach (our baselines). In these methods, the reliance on the online learning framework makes it impossible to eliminate any individual policy and thus cannot achieve sufficient sparsity. The online learner also requires extra hyper-parameters (learning rate, initialization etc.), which require extra tedious work to tune them. Inspired by Abernethy’s finding that standard Frank-Wolfe algorithm emerges as computing the equilibrium of a special zero-sum game, we proposed a novel reduction approach. We reduce the constrained problem to a distance minimization problem and solve it with FW type algorithms. Getting rid of the online learning framework enables elimination of unnecessary polices, and achieves better sparsity. Moreover, there is no need for extra hyper-parameters. (Modified and discussed in Sec 2)
>
> 4. Differences between our new variant of FW algorithm and Wolfe’s method.
> The objective function in Wolfe’s method is the norm function. However, the objective function in our problem is squared distance to a convex set, which unlike norm function, is not strongly convex. Moreover, affine minimizer with respect to a convex set is ill-defined. Therefore, we modified Wolfe’s method, and proposed a new variant of FW algorithm by adding a projection step in major cycle. In each major cycle, we minimize the distance toward the projected point. Since it is a new method, we analyze its complexity in Sec 4.4. Without the strongly convex property, our method is actually exponentially slower than Wolfe’s original method (linear convergence -> $O(1/\epsilon)$ convergence). (Modified and discussed in Sec 2, 4.2, 4.3)
>
> 5. Why not projection method?
> Because constructing a projection operator for $c(\Delta(\Pi))$ is non-trivial. For any given measurement vector, it is obscure how to modify a general RL algorithm to update the parameters such that the discounted expect measurement vector is closest to the given value. Therefore, projection-free methods are preferable for this task. In fact, by treating the set containing the target value as a convex set, the proposed C2RL method can be used to construct such a projection operator, and we leave it for further study. (Modified and discussed in Sec 4.1)
>
> 6. Reproducible concerns.
> We think a major obstacle for reproducibility is how the underlying RL algorithm is tuned. Therefore, in experiment part, we first provide results using an optimal oracle. Since this optimal oracle is build using Dijkstra to finds the shortest path under the current $\lambda$, it deterministically yields the optimal policy. Therefore, the performance of ApproPO and our C2RL can be compared fairly, by both calling the optimal oracle for 300 times. For the hyper-parameters, our method does not require any extra hyper-parameter so there is nothing to report. For completeness, parameters used in ApproPO, RCPO and the A2C agent (same as their original paper), are also included in the Appendix E of our latest paper.

---

### Decision · Program_Chairs · 2021-01-07
**Final Decision**

**Decision:**

Reject

**Comment:**

The paper builds on the prior work by Miryoosefi et al. (2019) that finds a feasible mixed policy under convex constraints through distance minimization over a simplex set. Instead of the primal-dual approach used in Miryoosefi et al. (2019), this paper proposes to apply Frank-Wolfe type algorithm (particularly, the minimum norm point algorithm) to promote sparsity of the mixed policy, while achieving the same complexity.

Despite the improvement on sparsity, the AC and some reviewers share two main concerns: (1) incremental novelty of the algorithm/theory, which basically follows from existing optimization work, (2) lack of (theoretical and numerical) justification of the significance of sparsity (especially given that the main computation costs come from projection and RL oracle).

Unfortunately, the paper lands just below borderline and cannot be accepted this time.